# Rapid nongenomic estrogen signaling controls alcohol drinking behavior in mice

Lia J. Zallar[1], Jean K. Rivera-Irizarry [2], Peter U. Hamor[3], Irena Pigulevskiy[1], Ana-Sofia Rico Rozo[3], Hajar Mehanna[3], Dezhi Liu [3], Jacqueline P. Welday [2], Rebecca Bender[3], Joseph J. Asfouri[3], Olivia B. Levine[2], Mary Jane Skelly[3,6], Colleen K. Hadley [4], Kristopher M. Fecteau [5], Scottie Nelson[3], John Miller[3], Pasha Ghazal [3,7], Peter Bellotti [3], Ashna Singh[2], Lauren V. Hollmer[1], David W. Erikson [5], Jacob Geri[3] & Kristen E. Pleil [1,2,3] ✉

Ovarian-derived estrogen can signal non-canonically at membrane-associated receptors in the brain to rapidly regulate neuronal function. Early alcohol drinking confers greater risk for alcohol use disorder in women than men, and binge alcohol drinking is correlated with high estrogen levels, but a causal role for estrogen in driving alcohol drinking has not been established. We found that female mice displayed greater binge alcohol drinking and reduced avoidance when estrogen was high during the estrous cycle than when it was low. The pro-drinking, but not anxiolytic, effect of high endogenous estrogen occurred via rapid signaling at membrane-associated estrogen receptor alpha in the bed nucleus of the stria terminalis, which promoted synaptic excitation of corticotropin-releasing factor neurons and facilitated their activity during alcohol drinking. Thus, this study demonstrates a rapid, nongenomic signaling mechanism for ovarian-derived estrogen in the brain controlling behavior in gonadally intact females.

Sex is a major regulator of behavior across species, but the numerous underlying neurobiological mechanisms that mediate these differences are not fully characterized[1]. One critical neuro-modulator of behavior is the sex steroid hormone estrogen (17β-estradiol; E2), which is produced in high volumes in the ovaries of females during the menstrual cycle in humans and estrous cycle in rodents[2]. E2 signaling is critical for the maintenance of numerous adaptive behaviors in both sexes, such as reproduction[3], learning and memory[4,5], and emotional behaviors[6], but it may also support maladaptive behaviors related to neuropsychiatric diseases[7–9]. E2 has been classically described to affect behavior through its engagement with nuclear estrogen receptors (ERs) to produce broad transcriptional changes via genomic mechanisms[1]. However, it can also act at membrane-associated ERs in the brain to rapidly modulate synaptic transmission and intrinsic excitability in brain regions including the hippocampus, hypothalamus, striatum, nucleus accumbens, and amygdala[5,10–16].

While a rapid signaling mechanism has been less explored for its role in E2 regulation of behavior, work in ovariectomized female rodents has established that E2 signals at membrane-associated ERs in the hippocampus to promote memory consolidation important for later recall using a modified E2 that is cell membrane-impermeable and thus can only signal at receptors at the membrane[17]. And, a host of research has demonstrated that E2 can rapidly modulate behavior in

[1]Pharmacology Graduate Program, Weill Cornell Graduate School of Medical Sciences, Weill Cornell Medicine, Cornell University, New York, NY, USA. [2]Neuroscience Graduate Program, Weill Cornell Graduate School of Medical Sciences, Weill Cornell Medicine, Cornell University, New York, NY, USA. [3]Department of Pharmacology, Weill Cornell Medicine, Cornell University, New York, NY, USA. [4]Weill Cornell/Rockefeller/Sloan Kettering Tri-institutional MD-PhD Program, New York, NY 10065, USA. [5]Endocrine Technologies Core, Oregon National Primate Research Center, Beaverton, OR, USA. [6]Present address: Psychology Department, Iona University, New Rochelle, NY, USA. [7]Present address: Department of Biosciences, COMSATS University Islamabad (CUI), Islamabad, Pakistan. ✉e-mail: krp2013@med.cornell.edu

gonadectomized rodents, implicating but not directly showing a rapid, membrane signaling mechanism. For example, acute administration of E2 or ER modulators can regulate behaviors including aggression[18,19] and reproductive/sexual behavior[20] in castrated male rodents, as well as learning and memory[21,22] and psychostimulant self-administration/ hyperlocomotion in ovariectomized female rodents[23–25]. In gonadally-intact males, pioneering work in birds has established that E2 (aromatized locally in birdsong brain regions from gonadal testosterone) rapidly regulates neuronal function to mediate birdsong and reproductive behaviors[26–30]; and, acute bioavailability of E2 is needed for male sexual behavior in intact male rodents[27,31]. Endogenous, ovarian-derived E2 may also be able to signal rapidly in the female brain to control behavior, but this has yet to be explored in gonadally-intact animals.

There is significant evidence that E2 is a key neuromodulator driving alcohol and drug-related behaviors[8,32–35], including excessive alcohol consumption and anxiety[9,35–37] that are primary risk factors for numerous psychiatric diseases including alcohol use disorder (AUD) and anxiety disorders[38–40]. Women exhibit an accelerated onset of, and a higher probability for developing, AUD with the same history of alcohol use as their male counterparts in a phenomenon known as the telescoping effect[41]; and, binge-like patterns of alcohol consumption are increasingly prevalent, especially in women[42,43]. Both binge alcohol drinking and related diseases emerge in women during puberty when ovarian-derived E2 and related hormones begin to fluctuate across the menstrual cycle, and many have reported positive associations between circulating concentration of E2 and both alcohol drinking behavior and anxiolysis[44–49]. It has been broadly reported that female rodents consume more alcohol than their male counterparts[50–52]. Ovarian hormones contribute to this phenomenon, as removal of the ovaries reduces alcohol drinking and increases avoidance behavior[9,35,53], and chronic E2 administration in ovariectomized females can restore high alcohol drinking and anxiolysis in some cases[9,54–59]. Foundational work has established that E2 signaling promotes alcohol drinking in intact females, as knockdown or degradation of ERα attenuates alcohol consumption[35,60]. However, a specific mechanism for the role of endogenous, ovarian-derived E2 in intact female alcohol drinking and avoidance behavior has not been established.

Here, we show that both alcohol drinking and avoidance behavior in female mice dynamically fluctuate across the estrous cycle in accordance with systemic E2 levels, contributing to observed sex differences in these behaviors. We show that the pro-drinking, but not anxiolytic, effects of ovarian E2 occur via rapid nongenomic signaling at membrane-associated ERα in the bed nucleus of the stria terminalis (BNST), one of the most sexually dimorphic brain regions that robustly expresses ERs[61,62]. Further, we find that BNST neurons that synthesize and release corticotropin releasing factor (CRF), a stress neuropeptide that has been highly linked to alcohol and anxiety-related behaviors[50,63–66] across mammalian species, are critical to the behavioral role of E2 via rapid E2-mediated increased excitatory synaptic transmission and enhancement of activity of BNST$^{CRF}$ neurons during motivated alcohol drinking behavior.

## Results

### Estrogen status across the estrous cycle dictates binge alcohol drinking and avoidance behavior

Prior studies have been unable to show dynamic modulation of alcohol drinking across the 4-5 day estrous cycle or a causal role for E2 in motivated alcohol drinking in gonadally intact female rodents, likely due to practical considerations including time of day of behavioral testing, potential disruption to estrous cyclicity by daily monitoring techniques, and the relatively transient peak of E2 during proestrus in rodents. Here, we validated and employed a minimally-invasive strategy to chronically monitor daily estrous cycle phase in intact female

mice in close temporal proximity to behavioral testing during the dark phase of the light:dark cycle when mice are most active and consume the most alcohol (Fig. 1a-d; Supplementary Fig. 1a-c). Two hours into the dark cycle each day, vaginal epithelial cells were collected via saline lavage and used to determine estrous cycle phase (Fig. 1a). Mice categorized as being in proestrus had higher ovarian expression of aromatase, the rate-limiting enzyme required for E2 synthesis (Fig. 1b), and higher blood plasma (Fig. 1c) and brain (Fig. 1d) E2 concentrations as measured by liquid chromatography tandem mass spectrometry (LC-MS/MS) than females in metestrus, confirming that mice categorized as being in proestrus using vaginal lavage-derived cytology were in a high ovarian E2 state (high E2) compared to those mice categorized as being in a low ovarian E2 state (low E2). Measures including estrous cycle length and the number of days in proestrus per estrous cycle were similar when lavage was performed in the light and dark phases of the light:dark cycle, and normal estrous cyclicity remained consistent across many weeks of daily monitoring (Supplementary Fig. 1a-c). Therefore, we were able to use this vaginal cytology tracking strategy as an accurate, minimally-invasive proxy measure of real-time E2 status in intact female mice during behavioral testing in the dark phase without disrupting their natural estrous cyclicity and hormonal milieu.

We first employed this strategy to assess the effects of high E2 status during the estrous cycle on binge alcohol drinking using the standard drinking in the dark (DID) paradigm in which mice receive limited access (2–4 h/day, 4 consecutive days per cycle) to 20% EtOH instead of their home cage water bottle (Fig. 1e). Across weekly cycles of EtOH DID, individual females consumed more alcohol when they were in a high E2 state in proestrus compared to when they were in other estrous cycle phases, with alcohol consumption during low E2 days increasing until it converged with the high E2 level by cycle six (Fig. 1f), demonstrating that individual females consume more alcohol when in a high ovarian E2 state than low ovarian E2 state. There was no significant difference in alcohol consumption between low E2 females and males, suggesting that ovarian E2 contributes to the higher consumption in females than males that we and others have previously reported[50,52,67]. Intriguingly, when females were specifically given their first access to alcohol in a high ovarian E2 state in proestrus, they consumed more alcohol than conspecifics given first access in a low E2 state (Fig. 1g), suggesting that prior experience with alcohol was not required for the pro-drinking effect of E2 status on binge drinking behavior. Further, there was no correlation between the cumulative number of alcohol drinking sessions in a high E2 state and alcohol consumption during subsequent low E2 sessions within individual mice across the six EtOH DID cycles (Supplementary Fig. 1d), suggesting that the acute stimulatory effect of high E2 state on alcohol drinking was not directly related to the gradual escalation in low E2 alcohol consumption across DID cycles. In contrast to alcohol intake, estrous cycle did not affect consumption of 1% sucrose in a matched DID paradigm (sucrose DID; Fig. 1h,i). Females showed similar sucrose intake when in high and low E2 states and compared to males across cycles of sucrose DID (Fig. 1h) and during their first access to sucrose (Fig. 1i), suggesting the effects of ovarian E2 status on alcohol drinking are not generalized to palatable/rewarding substances or appetitive regulation.

High volume alcohol consumption is often associated with dysregulation of anxiety[68], and anxiety behavior itself is also sensitive to ovarian-derived E2 status in humans[46,49,69] and rodents[36,37,70]. To provide insight into whether ovarian E2 dynamically regulates avoidance behavior in intact female mice, we first performed a battery of avoidance behavior assays one week apart. Females in a high E2 state displayed an anxiolytic phenotype compared to conspecifics in a low E2 state and males, with greater percent time spent and distance in the center of the open field (OF), light side of the light:dark box (LDB), and open arms of the elevated plus maze (EPM), without any alteration in

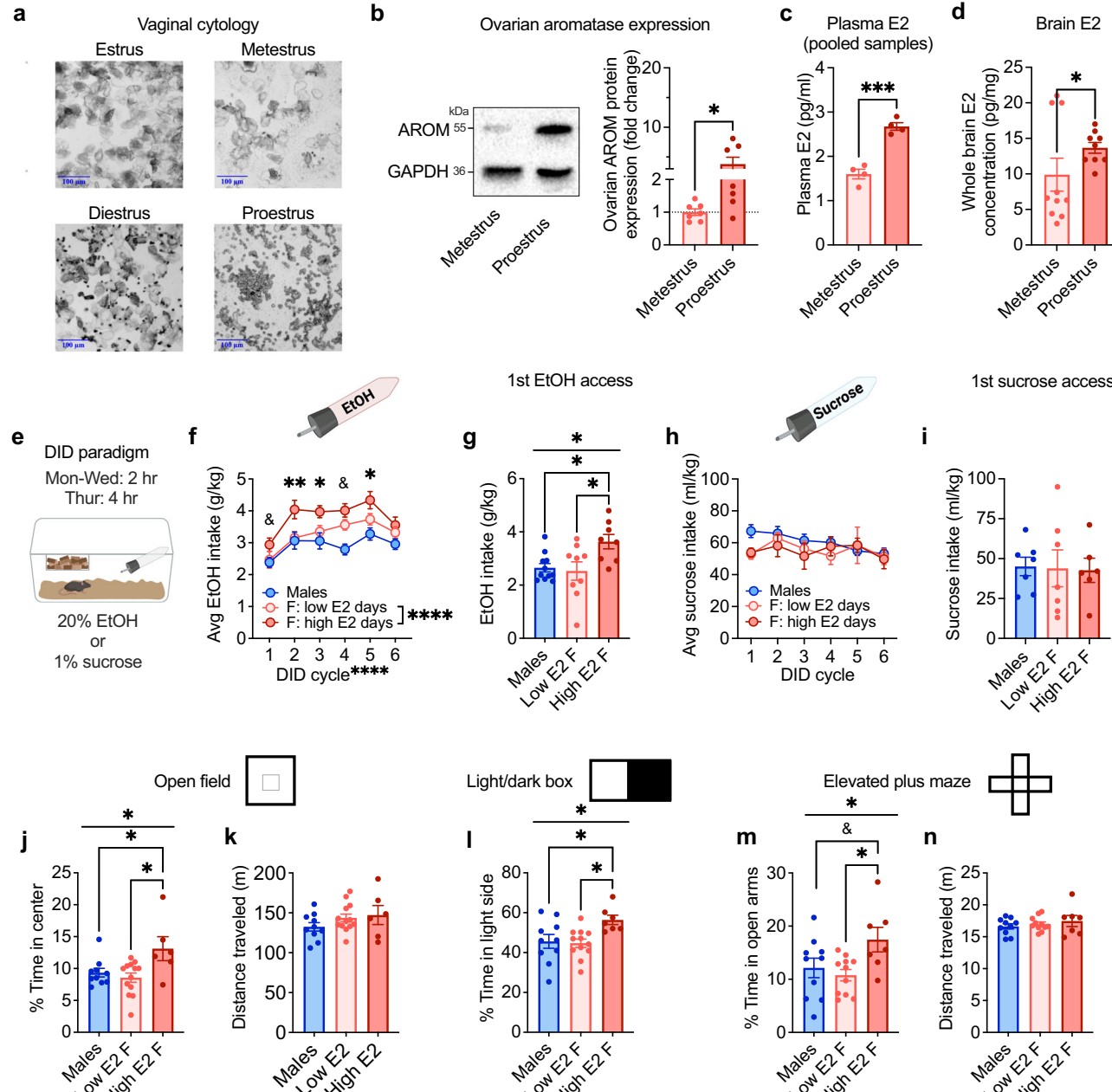

**Fig. 1 | Binge alcohol drinking and avoidance behavior fluctuate with E2 status across the estrous cycle. a** Representative images of vaginal epithelial cells across the estrous cycle. **b** Ovarian aromatase (AROM) protein expression in female mice categorized as proestrus compared to metestrus status females (N's = metestrus 7, proestrus 7). **c** Plasma E2 concentrations in proestrus and metestrus status females (N's = metestrus 4, proestrus 4; each N represents a pooled sample from five mice). **d** Whole-brain E2 concentrations in proestrus and metestrus status females (N's = metestrus 10, proestrus 9). **e** Drinking in the Dark (DID) binge drinking paradigm (Biorender license: VB27JLIIQ2). **f** Average 2-h consumption of EtOH across cycles of DID (N's = 27 F, 7 M; Biorender license: RP27GVXCTW). **g** First EtOH access (N's = high E2 8, low E2 9, M 11). **h** Average 2-h consumption of sucrose across cycles of DID (N's = 15 F, 15 M; Biorender license: DD27GVXH5F). **i** First sucrose consumption (N's = high E2 6, low E2 7, M 7). **j** % Time spent in the center of the open field (OF; N's = high E2 6, low E2 13, M 10). **k** Distance traveled in the OF (N's = high E2 6, low E2 13, M 10). **l** % Time on the light side of the light:dark box (LDB; N's = high E2 7, low E2 12, M 10). **m** % Time in open arms of the elevated plus maze (EPM; N's = high E2 7, low E2 11, M 10). **n** Distance traveled in the EPM (N's = high E2 7, low E2 11, M 10). *$P < 0.05$, **$P < 0.01$, ***$P < 0.001$, ****$P < 0.0001$, &$P < 0.10$ for unpaired t-tests, mixed effects analysis main effects, one way ANOVA main effects of group, and post hoc t-tests paired t-tests with H-S corrections between high and low E2 days in **f** and post hoc unpaired t-tests with H-S corrections for other panels, as indicated. Data are presented as mean values +/−SEM. Detailed statistics provided in Supplemental Table 1. Source data are provided as a Source Data file.

their overall locomotor behavior (Fig. 1j–n; Supplementary Fig. 1e, f). Individual females' behavior varied across the battery according to their estrous cycle stage on each test day, suggesting that ovarian E2 is a critical real-time regulator of avoidance behavior at the individual level, even when mice have had prior experiences in experimentally-induced anxiogenic contexts.

## High ovarian estrogen state enhances excitatory synaptic transmission in BNST[CRF] neurons and their pro-alcohol drinking role

Data from human patients with alcohol use disorder and anxiety disorders show that the BNST is an important hub of neurocircuitry altered in these disease states[71], and the BNST is critical in the

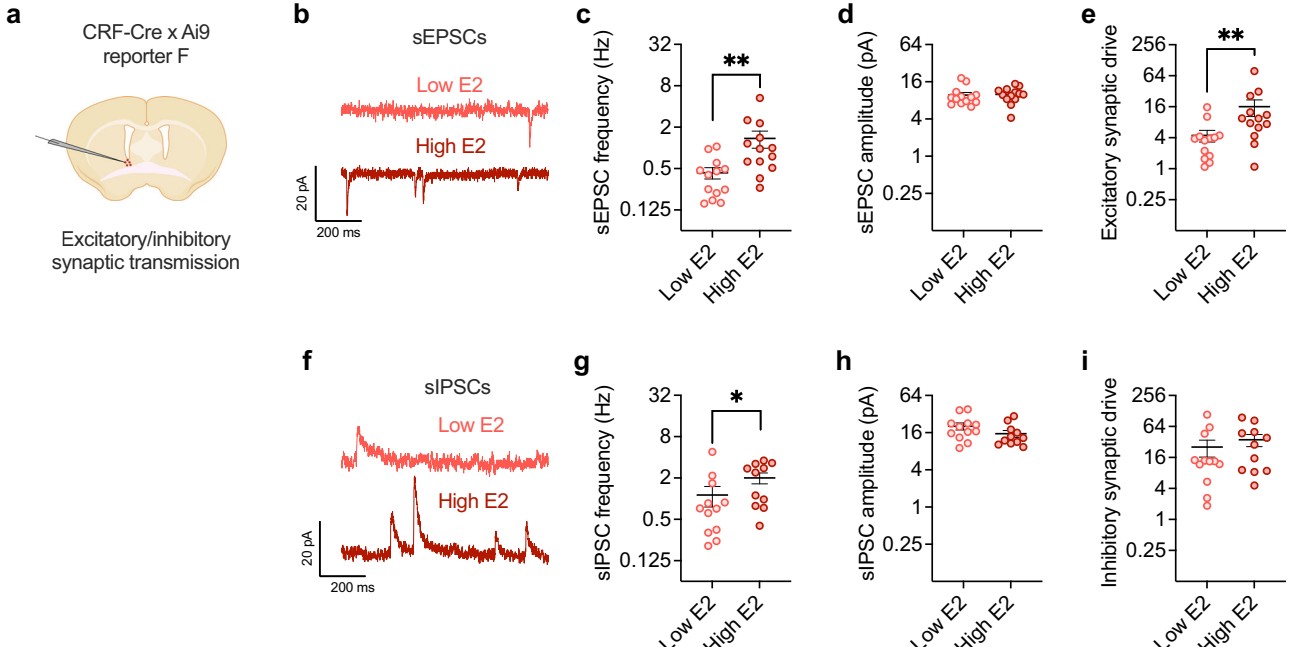

**Fig. 2 | BNST^CRF neurons are sensitive to ovarian estrogen status.** Spontaneous synaptic transmission in BNST^CRF neurons as measured in slice electrophysiology recordings from intact CRF-CrexAi9 reporter females in low vs. high ovarian E2 states, with schematic in **a** (Biorender license: QD27GVXM7M). **b** Representative traces of spontaneous excitatory postsynaptic currents (sEPSCs). **c** The frequency of sEPSCs in BNST^CRF neurons from high and low ovarian E2 status females (N's = 6 low E2, 13 cells; 7 high E2, 13 cells; **d** and **e** are the same cells). **d** sEPSC amplitude in high and low ovarian E2 status female BNST^CRF neurons. **e**) Excitatory synaptic drive onto BNST^CRF neurons (frequency x amplitude) high and low ovarian E2 status females. **f** Representative traces of spontaneous inhibitory postsynaptic currents (sIPSCs). sIPSC frequency in BNST^CRF neurons high and low ovarian E2 status females (**g**), amplitude (**h**), and synaptic drive (**i**; N's = 6 low E2, 12 cells; 7 high E2, 11 cells). *$P < 0.05$, **$P < 0.01$ for unpaired t-tests between low E2 and. high E2 groups. Data are presented as mean values +/−SEM. Detailed statistics provided in Supplemental Table 1. Source data are provided as a Source Data file.

regulation of binge alcohol drinking and avoidance behavior in rodents[50,65]. While the BNST richly synthesizes and releases numerous neuropeptides, the stress neuropeptide CRF has been most robustly linked to alcohol consumption[72,73]. A major subpopulation of CRF expressing BNST neurons is critical for driving alcohol drinking in both sexes and avoidance behavior in males (females not examined)[50], and project to numerous regions associated with stress, reward, and alcohol use[74,75]. We recently reported that BNST^CRF neurons in females display greater basal excitability and glutamatergic synaptic transmission than in males, which may be related to their higher alcohol drinking, and that they are required for binge alcohol drinking in both sexes[50]. Here, using a multiplexed chemogenetic manipulation approach (Supplementary Fig. 2), we replicated our previous finding that the activity of BNST^CRF neurons promote females' binge alcohol drinking (Supplementary Fig. 2c), and we further found that it surprisingly does not modulate avoidance behavior in females (Supplementary Fig. 2d) as we have previously found in males[50]. During slice electrophysiology recordings (Fig. 2), we found that BNST^CRF neurons from females in a high ovarian E2 state had a higher frequency, but not amplitude, of spontaneous excitatory postsynaptic currents (sEPSCs) than those from females in a low ovarian E2 state (Fig. 2c, d), leading to greater excitatory synaptic drive (a composite metric that takes into account both the frequency and amplitude of PSC events[50]; Fig. 2e). BNST^CRF neurons from high E2 females also displayed higher frequency, but not amplitude, of spontaneous inhibitory postsynaptic currents (sIPSCs; Fig. 2g, h), which was insufficient to enhance inhibitory synaptic drive (Fig. 2i). These results suggest that high ovarian-derived E2 may enhance the excitatory synaptic transmission onto BNST^CRF neurons, promoting alcohol drinking in females that contributes to their greater alcohol consumption than males.

To examine whether ovarian E2 status regulates BNST^CRF neuron excitation in vivo during alcohol drinking behavior, we used the

calcium biosensor GCaMP6s in BNST^CRF neurons as a proxy measure of population-level neuronal activity during a modified EtOH DID task. Mice received water access before (W1) and after (W2) the 20% EtOH access during each day of DID (Fig. 3; Supplementary Fig. 3), and lickometer tracking of drinking behavior was used to characterize the temporal kinetics of and relationship between water/alcohol drinking, BNST^CRF neuron activity, and ovarian E2 status. We found that mice displayed greater motivated.

Alcohol drinking, defined by more licking bouts (>1 s duration), when in a high E2 than low E2 state specifically during the first 30 min of EtOH access, including greater time spent drinking and higher average bout duration (Fig. 3c, e, f). However, the total number of bouts, licking behavior, and fluid consumption across the entire access period did not differ between high E2 and low E2 drinking days (Fig. 3d; Supplementary Fig. 3d). These data confirmed that the effects of ovarian E2 state on drinking within individual mice and across days of drinking were specific to alcohol rather than general fluid intake, and they point to the first 30 min of EtOH access as the most sensitive to E2 state-mediated regulation of alcohol drinking. Examination of BNST^CRF GCaMP activity during alcohol and water drinking showed that it ramped up at the onset of drinking bouts of EtOH, but not water, on both high E2 and low E2 drinking days (Fig. 3h, i), and the time spent drinking alcohol was positively correlated with average GCaMP signal during the first 30 min of EtOH access (Fig. 3j), suggesting that high BNST^CRF activation was associated with greater alcohol drinking.

Because females drank more alcohol early into access on high E2 days than low E2 days and consumption was correlated with BNST^CRF calcium activity, we posited that E2-dependent changes to GCaMP activity were related with behavioral changes in motivated drinking specifically across the transition between water and EtOH access. Indeed, we found that there was an increase in both the number

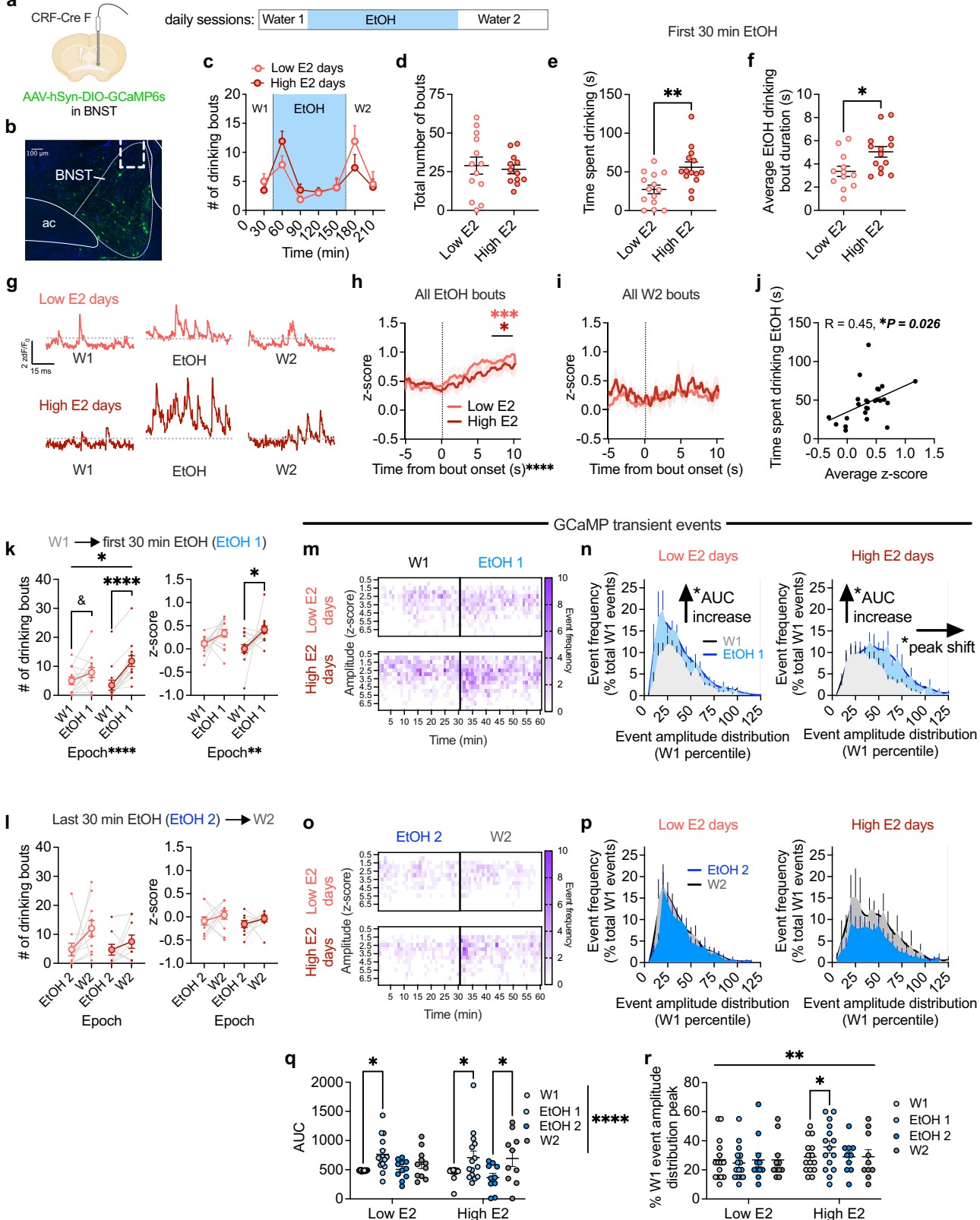

of motivated drinking bouts and overall BNST$^{CRF}$ neuron GCaMP activity across the W1 epoch and the first 30 min of EtOH (EtOH 1) on high E2 but not low E2 drinking days (Fig. 3k), while neither measure differed between the last 30 min of EtOH (EtOH 2) and second water (W2) epochs in either E2 state (Fig. 3l), again suggesting that E2 state-mediated changes in overall BNST$^{CRF}$ neuron activity were related to alcohol drinking in particular. Analysis of the calcium transient events

resulting from BNST$^{CRF}$ neuron activity showed that the frequency of events increased across the W1→EtOH 1 epoch transition on both high E2 and low E2 days and across the EtOH 2→W2 epoch transitions on high E2 days, indicated by a greater area under the curve for the distribution of epoch event data normalized to W1 on each day (Fig. 3m–q; Supplementary Fig. 3e, h). Further, specific examination of the temporal kinetics of event frequency, independent of amplitude,

**Fig. 3 | Ovarian E2 state modulates the activity of BNST^CRF neurons during motivated alcohol drinking.** Fiber photometry recordings of GCaMP6s in BNST^CRF neurons, with schematic of unilateral viral injection and optical fiber cannula placement (**a**; Biorender license: QP27GVY1EL) and representative image of GCaMP expression/fiber placement (**b**; ac: anterior commissure). **c** Modified EtOH DID timeline with water access given before (W1) and after (W2) the 2-h EtOH access period, and drinking bouts time course on high and low E2 days. **d** Total drinking bouts across the session (EtOH + W2; N's=5 for all sub figures, 13 low E2, 12 high E2). Total time spent displaying motivated EtOH drinking (time in bout; **e**; 14 low E2, 14 high E2) and average EtOH bout duration (**f**; 12 low E2, 14 high E2). **g** Representative traces of GCaMP signal from one mouse on a low E2 and high E2 day. GCaMP signal during high and low E2 days 10 seconds following bout onset for EtOH (**h**) and water (**i**; 13 low E2, 15 high E2 for both). **j** Correlation between GCaMP signal and time spent motivated drinking during the first 30 min of EtOH (25 days). **k** The number of drinking bouts (13 low E2, 13 high E2) and average GCaMP signal (14 low E2, 14 high E2) during the first 30 min of EtOH (EtOH 1) compared to W1. **l** Bout number (11 low E2, 9 high E2) and GCaMP signal (11 low E2, 9 high E2) in the first 30 min of W2 compared to the last 30 min of EtOH (EtOH 2) on high and low E2 days. **m** Representative heat map of the frequency of GCaMP transient events across amplitude bins for a single mouse across W1 and EtOH 1 on low vs. high E2 days. **n** Frequency distribution of event amplitudes normalized to the amplitude distribution for W1 within each day during W1 and EtOH 1 epochs. **o** Representative heat map of the frequency of GCaMP events across amplitude bins across EtOH 2 and W2 for the same mouse as in **l**. **p** Frequency distribution of event amplitudes normalized to the amplitude distribution for W1 during EtOH 2 and W2. **q** Area under the curve (AUC) of GCaMP event distributions shown in **n** and **p** (W1 and EtOH 1: 15 low E2, 15 high E2; EtOH 2 and W2: 11 low E2, 10 high E2). **r** The peak of the event amplitude distributions shown in **n** and **p** (same N's depicted in **q**). \*$P < 0.05$, \*\*$P < 0.01$, \*\*\*$P < 0.001$, \*\*\*\*$P < 0.0001$, ^&$P < 0.10$ for unpaired t-tests, 2xANOVA main effects and interactions and their post hoc paired t-tests with H-S corrections, and Pearson's correlation. Data are presented as mean values +/−SEM. Detailed statistics are provided in Supplemental Table 1. Source data are provided as a Source Data file.

showed that it briefly (first 5 min) increased upon the presentation of a new bottle (containing either EtOH or water) regardless of E2 state (Supplementary Fig. 3f, i), suggesting that BNST^CRF neuron activity is sensitive to the bottle cue and/or novelty associated with the bottle change, a known stimulator of consumption[76]. In contrast, event amplitude increased across the W1→EtOH 1 epoch transition, but not the EtOH 2→W2 epoch transition, and specifically on high E2 drinking days (Fig. 3m–p, r; Supplementary Fig. 3f, i); this was indicated by a significant rightward shift for the frequency distribution of epoch event amplitude for the EtOH 1 epoch on high E2 drinking days (Fig. 3m–p, r), demonstrating that the added events during EtOH 1 specifically on high E2 days were larger in amplitude. This increase in event amplitude emerged at the onset of EtOH 1 and persisted throughout (Fig. 3m; Supplementary Fig. 3f, i). Thus, in both high and low E2 states, the frequency of calcium transient events increased during alcohol consumption compared to W1, as indicated by an increase in AUC (Fig. 3n, q). However, the distributions show that in a low E2 state, these added events during EtOH 1 are of the same amplitude, while in a high E2 state the EtOH 1 added events have a greater amplitude compared to W1, indicated by the rightward shift in the amplitude distribution (Fig. 3n, r). These results suggest temporal summation of BNST^CRF neuron activity due to higher frequency action potential firing within the activated BNST^CRF population and/or greater coordinated activity across the population[77] during early alcohol access when drinking in a high E2 state. As female BNST^CRF neurons project broadly across the extended amygdala, hypothalamus, thalamus, midbrain, and brainstem (Supplementary Fig. 4), a high E2 state may prime the increase in activity of specific BNST^CRF projection populations or recruitment of additional projections during alcohol access.

We also examined the E2 state-dependence of BNST^CRF neuron activity during avoidance behavior (Supplementary Fig. 3j, k). Surprisingly, while BNST^CRF neuron GCaMP activity was sensitive to the anxiogenic context in both the EPM and OF assays, with highest activity in the open arms of the EPM and center of the OF and lowest activity in the closed arms of the EPM and corners of the OF, it did not differ between mice on high or low ovarian E2 status days. Altogether, these results suggest that high ovarian E2 state-dependent excitation of BNST^CRF neurons contributes to the pro-alcohol drinking, but not anxiolytic, phenotype observed in proestrus.

### Rapid nongenomic E2 signaling in the BNST promotes binge alcohol drinking and increases excitatory synaptic transmission in BNST^CRF neurons

We next assessed the timescale on which ovarian E2 can regulate binge alcohol drinking and avoidance behaviors (Fig. 4, Supplementary Fig. 5). We found that systemic administration of the aromatase inhibitor letrozole (LET) to acutely block E2 synthesis (and thus acutely reduce E2 bioavailability) dose-dependently suppressed binge alcohol consumption in females in a high ovarian E2 state (Fig. 4a, b) but not in a low ovarian E2 state (Supplementary Fig. 5a, b). Acute LET administration had no effect on binge sucrose consumption in either high or low ovarian E2 females (Supplementary Fig. 5c), suggesting that E2's role in promoting alcohol drinking is not due to modulation of general appetitive behavior or reward sensitivity.

In contrast to its effects on alcohol consumption, systemic administration of LET did not attenuate high E2-mediated anxiolysis in the EPM (Fig. 4c; Supplementary Fig. 5d). And, acute intra-BNST infusion of LET did not alter binge alcohol consumption in high ovarian E2 status females (Fig. 4d, e), suggesting that the source of E2 synthesis driving the pro-binge drinking phenotype is not the BNST. Together, these results demonstrate that acute bioavailability and signaling of ovarian-derived E2 in the brain are necessary for the pro alcohol drinking, but not anxiolytic, effects of high ovarian E2 status. As E2 availability in real time was required for the pro-alcohol drinking effects of ovarian E2 status, we next sought to understand whether the BNST is a critical site for the behavioral effects of rapid E2 signaling, likely via nongenomic signaling at the cell membrane (Fig. 4f–i). We found that acute intra-BNST infusion of E2 (20 pg/side) in low E2 status females 10 min prior to behavioral testing rapidly enhanced binge alcohol consumption measured at one and two hours (Fig. 4g). We further found that intra-BNST infusion of a purified BSA-conjugated E2 that cannot cross the cell membrane and thus can only signal at the membrane (membrane-only E2, mE2), also rapidly increased binge alcohol drinking in low ovarian E2 status females at 1 h (Fig. 4h). In contrast, acute intra-BNST infusion of E2 had no effect on avoidance behavior on the EPM (Fig. 4i). These results suggest that the BNST is a site for the rapid E2 signaling effects of high ovarian E2 status driving binge alcohol drinking behavior through nongenomic E2 signaling at membrane-associated ERs; further, they suggest that the E2-induced transcriptional program of high ovarian E2 status is not required for these pro-drinking effects but may be required for the anxiolytic effects of high ovarian E2 status.

Because BNST^CRF neuron activation and acute E2 signaling in the BNST both rapidly enhanced binge alcohol drinking, we tested whether E2 rapidly promotes excitatory synaptic transmission in BNST^CRF neurons in slice electrophysiology recordings (Fig. 4j–m). We found that bath application of E2 across a range of doses (0.1 nM-1 μM) and mE2 (100 nM) robustly increased the frequency of sEPSCs in a large proportion of BNST^CRF neurons in low ovarian E2 females (Fig. 4k, l; Supplementary Fig. 6a, b). sEPSC frequency was modulated by rapid E2 signaling to a greater degree than sEPSC amplitude, and E2-mediated changes in sEPSC frequency and amplitude were independent of one another (Fig. 4l, m; Supplementary Fig. 6c, d). The

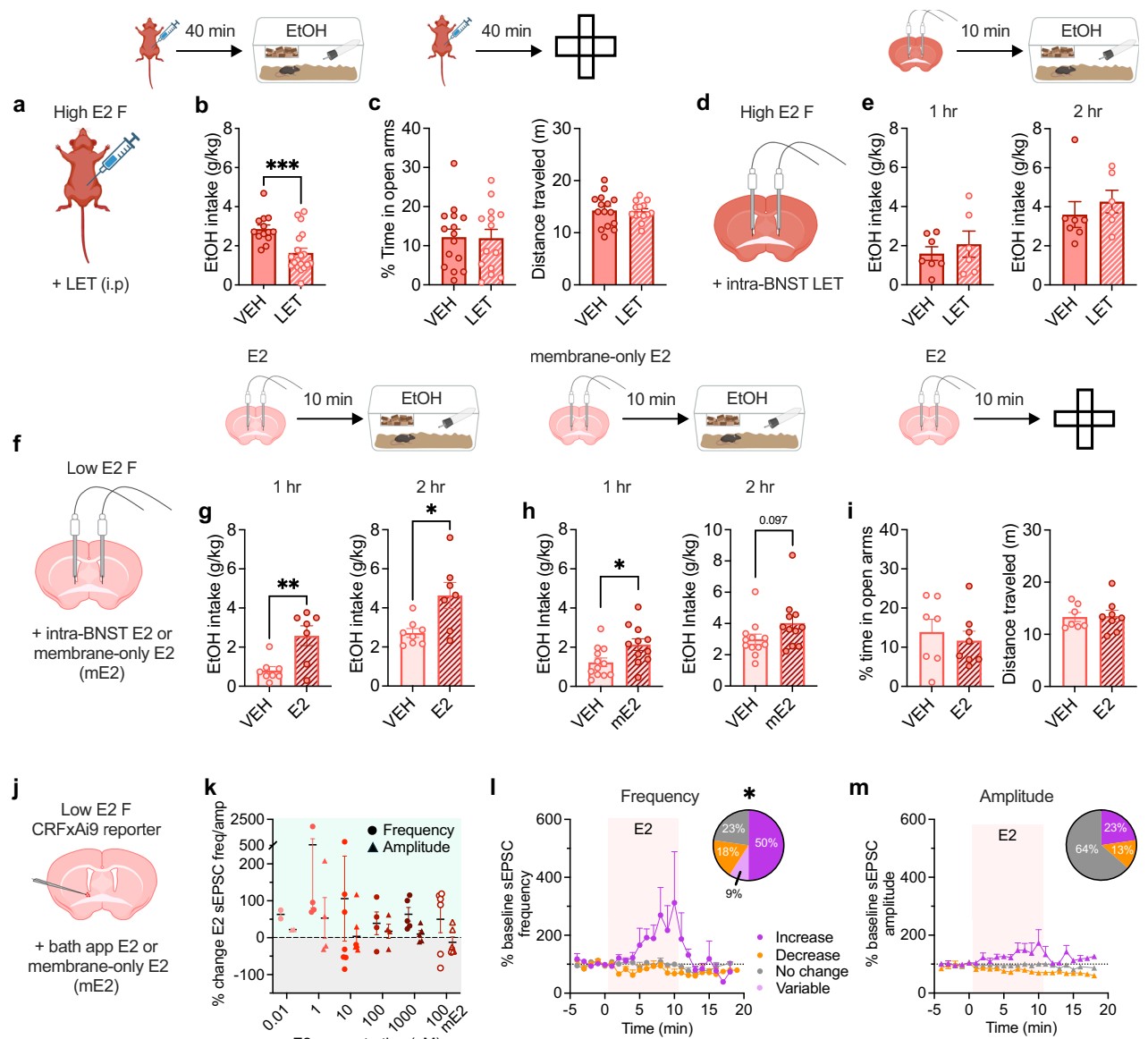

**Fig. 4 | Rapid E2 signaling in the BNST recapitulates the pro-drinking but not anxiolytic effects of ovarian E2 and modulates BNST^CRF neurons. a** Systemic E2 synthesis inhibition in high ovarian E2 mice using the aromatase inhibitor letrozole (LET, 10 mg/kg; Biorender license: UM27GVZ81B). **b** Effects of LET administration on EtOH consumption (N's = 13 saline VEH, 19 LET; Biorender license: VB27JLIIQ2). **c** Effects of LET administration on elevated plus maze (EPM) % time spent in the open arms (left) or distance traveled (right; N's = 15 VEH, 14 LET). **d** Intra-BNST infusion of LET (1 μg in 200 nl/side) in high E2 status females. **e** Effects of intra-BNST LET on EtOH consumption (N's = 7 VEH, 6 LET). **f** Intra-BNST infusion of E2 (20 pg in 200 nl/side) or membrane-impermeable E2 (mE2, 55 pg in 200 nl/side) in low E2 status females (Biorender license: BF27GVZGF6). **g** Effects of intra-BNST E2 on EtOH consumption (N's = 8 VEH, 7 E2). **h** Effects of intra-BNST mE2 on EtOH consumption (N's = 12 VEH, 11 mE2). **i** Effects of intra-BNST E2 on EPM % time spent in the open arms (left) or distance traveled (right; N's = 7 VEH, 8 E2). **j** Effects

of bath application of E2/mE2 on excitatory synaptic transmission in BNST^CRF neurons during slice electrophysiology recordings in low ovarian E2 status female CRF-CrexAi9 reporters (Biorender license: JF27GVZUJE). **k** Spontaneous excitatory postsynaptic currents (sEPSCs) maximum delta from % baseline during E2/mE2 wash on (E2 nM: 0.01: N = 2, 2 cells; 1: N = 4, 4 cells; 10: N = 5, 7 cells; 100: N = 4, 4 cells; 1000: N = 3, 5 cells; 100 nM mE2: N = 3, 6 cells; l and m are the same cells). **l** Time course of BNST^CRF neurons that displayed increase/decrease/no change in sEPSC frequency and amplitude (**m**) % change from baseline during E2 application and proportion of responding categories (pie charts). *P < 0.05, **P < 0.01, ***P < 0.001 for unpaired t-tests between VEH vs. LET treatment and VEH vs E2/mE2 treatment; Fisher's exact test frequency vs amplitude; post hoc t-tests with H-S corrections as indicated. Data are presented as mean values +/−SEM. Detailed statistics are provided in Supplementary Table 1. Source data are provided as a Source Data file.

enhancement of sEPSC frequency by bath-applied E2 (Fig. 4k, l) recapitulated the effects observed in high E2 status females (Fig. 2i, j), peaking at an average of 355% of baseline during the 10 min E2 application period and returning to baseline during washout (Fig. 4l). These data suggest that E2 signals via a rapid nongenomic mechanism to promote glutamatergic drive of a large subpopulation of BNST^CRF neurons, perhaps defined by projection population/circuit contribution (Supplementary Fig. 4), and this mechanism may

contribute to the pro-alcohol drinking role of rapid E2 signaling in the BNST.

## ERα mediates the effects of rapid E2 signaling in the BNST

To investigate the receptor(s) mediating E2 drive of alcohol drinking, we examined the expression of ERs in BNST circuits. Analysis of single nucleus RNA sequencing (snRNA-seq) dataset (GEO: GSE126836)[78] showed that ERα (*Esr1*) was robustly expressed in the BNST (29.3%),

while ERβ (*Esr2*) was sparsely expressed (6.7%; Fig. 5a–c), and the membrane-only receptor GPER (*Gper1*) was nearly undetected (0.3%; not shown). ERα was enriched in subpopulations of BNST cells that modulate glutamatergic transmission at BNST$^{CRF}$ synapses, including CRF neurons themselves, VGLUT2+ (*Slc17a6*) glutamatergic BNST neurons, and astrocytes (*Gfap*) that regulate synaptic glutamate levels (Fig. 5a–c, Supplementary Fig. 7a–d). We confirmed ERα and ERβ expression profiles in the BNST using high-sensitivity RNAscope fluorescence in situ hybridization (FISH). ERα was robustly expressed and ERβ sparsely expressed in CRF+ neurons (Fig. 5d) and VGLUT2+ neurons (Fig. 5e) in the BNST. We then assessed whether ERα mediates rapid excitatory synaptic transmission in BNST$^{CRF}$ neurons of high E2 females via bath application of the ERα antagonist methyl-piperidino-pyrazole (MPP; 3 μM; Fig. 5f). MPP rapidly reduced sEPSC frequency in the majority of cells (71.4%) by an average of 51.5% of baseline, with no significant effect on sEPSC amplitude (Fig. 5g, h, Supplementary Fig. 7e, f). In contrast, MPP reduced sEPSC frequency in only 28.6% of low E2 status BNST$^{CRF}$ neurons (Supplementary Fig. 7g–i). This suggests that there is E2 tone in the BNST of high ovarian E2 females that contributes to greater sEPSC frequency via rapid signaling at ERα; the rapid nature of the effect of ERα antagonism points to E2-ERα signaling at the membrane mediating the effects of ovarian E2 modulation. Finally, we probed whether this rapid E2 signaling in the BNST of high E2 females was required for the behavioral effects of high ovarian E2 state using acute intra-BNST administration of the ERα antagonist MPP (10 μM), the ERβ antagonist 4-[2-phenyl-5,7-bis(trifluoromethyl) pyrazolo[1,5-a]pyrimidin-3-yl]phenol (PHTPP; 10 μM), or their control vehicles 10 min prior to behavioral testing (Fig. 5i). We found that acutely blocking ERα, but not ERβ, signaling in the BNST abolished the pro-alcohol drinking effects of high ovarian E2 status (Fig. 5j), with no effect of intra-BNST MPP in low ovarian E2 status females (Supplementary Fig. 7j, k). Neither intra-BNST ERα nor ERβ antagonism affected avoidance behavior in high ovarian E2 status females (Fig. 5k, l). Altogether, our results suggest that rapid E2 signaling at membrane-bound ERα mediates the effects of E2 on BNST$^{CRF}$ neuron excitation and is required for the pro-alcohol drinking effects of high ovarian E2 status in intact female mice. In contrast, basal anxiety state that is sensitive to ovarian E2 status is unaffected by acute E2 manipulations; rather, the transcriptional program of high ovarian E2 status may be necessary for E2's anxiolytic effects.

## Discussion

Here, we describe a critical role for rapid E2 signaling in driving binge drinking behavior in female mice. We found that estrous cycle status dictates behavioral phenotypes in female mice, as females displayed greater binge alcohol drinking and reduced avoidance behavior when they were in a high ovarian E2 state than when they were in a low ovarian E2 state (Fig. 1). However, the mechanisms driving these effects diverged, as rapid E2 signaling was required for the pro-alcohol drinking, but not anxiolytic, effects of high ovarian E2 (Figs. 4 and 5). Further, BNST$^{CRF}$ neuron activity was enhanced in a high E2 state during early alcohol access to promote motivated alcohol consumption, while it was neither necessary for nor sensitive to ovarian E2 state during avoidance behavior (Figs. 2 and 3, Supplementary Fig. 3). Intra-BNST E2 signaling, including that which could only signal at the cell membrane (mE2), rapidly promoted excitatory synaptic transmission in BNST$^{CRF}$ neurons and binge alcohol drinking in low E2 females (Fig. 4). And, ERα, which was densely and preferentially expressed within BNST$^{CRF}$ glutamatergic circuit components, mediated both the pro-drinking effects of high ovarian E2 state and its rapid enhancement of synaptic excitation of BNST$^{CRF}$ neurons (Fig. 5), as E2 increased these measures in low E2 mice and ERα antagonism reduced these measures in high E2, but not low E2, mice. Altogether, these results suggest that nongenomic E2 signaling at membrane-associated ERα in the BNST mediates the pro-drinking effects of ovarian E2 via excitation of BNST$^{CRF}$ neurons.

There is significant evidence of an association between E2 and alcohol consumption in humans[46,79], but historically the relationship between drinking and menstrual status has proven to be difficult to parse[80]. Careful monitoring of menstrual cycle stage and ovarian hormone status in nonhuman primates has been able to demonstrate that alcohol intake is modulated by fluctuating ovarian hormones across the menstrual cycle[81], and a recent meta-analysis reported that alcohol consumption in women is indeed elevated during ovulation when ovarian-derived E2 concentrations are high[82] Previous work has shown that psychostimulant[32] and opioid-related[83] behaviors are modulated across the estrous cycle in rodents, however demonstrating estrous cycle modulation of alcohol drinking has proven to be challenging[9,51,84,85]. Nonetheless, many studies report that ovariectomy reduces alcohol intake[9,86] and E2 can enhance drinking in ovariectomized[9,56] and intact animals[57], suggesting a role for fluctuating E2 across the intact female rodent estrous cycle in alcohol drinking. Here, we validated a precise and accurate method for longitudinal monitoring of ovarian E2 status in intact female mice during behavior that allowed females to cycle normally. Using this, we provide the first evidence that fluctuating, rapidly acting ovarian-derived E2 is the source for the identified estrous cycle-dependent pro-drinking phenotype. In addition, we provide mechanistic insight into the role of ovarian E2 in alcohol drinking, showing that E2 signaling at membrane-associated ERα in the BNST drives binge alcohol drinking behavior. While intra-BNST membrane-only E2 (mE2) did increase alcohol consumption at the 1-h timepoint, it should be noted that it did not have a robust effect at 2 h, in contrast to intra-BNST free E2 (Fig. 4). This may suggest that while rapid E2 signaling via membrane bound ERs is critical for binge drinking, E2 may act through multiple mechanisms to regulate behavior. As males also have robust expression of ERs in the BNST that mediate behavior[19], locally aromatized E2 in the BNST may be able to regulate binge alcohol drinking in males through a similar E2-dependent mechanism. This may be similar to what has been described for rapid E2 synthesis and signaling modulation of birdsong and reproduction in birdsong and hypothalamic nuclei in birds[27,30,87].

Here, we find that rapid signaling of ovarian E2 at membrane-associated receptors promotes excitatory synaptic transmission in BNST$^{CRF}$ neurons that drive motivated alcohol drinking. Although the link between rapid E2 signaling and acute behavioral control has not previously been described in gonadally-intact female rodents, there is a rich literature detailing signaling mechanisms of membrane-associated ER signaling in ex vivo and in vitro experiments[87–93]. This effect on synaptic excitation is similar to studies in the hippocampus and hypothalamus showing E2-mediated enhancement of glutamatergic mechanisms including glutamate release, dendritic spine formation, and long term potentiation[21,94–96]. The current reports on the role of rapid E2 in ovariectomized females on behavioral control show that social memory is dependent on rapid-E2 mediated ERK and PI3K actions in the hippocampus[22], and E2-mediated enhancement of cocaine self-administration requires mGluR5 activity[97], indicating the importance of excitatory synaptic transmission in E2's behavioral roles. Future work is necessary to fully elucidate the signaling mechanism underlying the role of rapid E2 in driving BNST$^{CRF}$ neuron activity to promote motivated alcohol drinking. Further, exogenous E2 and pharmacological activation of ERα rapidly increase alcohol-induced excitation of VTA dopamine neurons[60,98]. As the VTA is a primary target of BNST$^{CRF}$ neurons[75], and we found that a large subpopulation of BNST$^{CRF}$ neurons were sensitive to rapid E2 modulation, it is possible that E2 increases the activation of BNST$^{CRF}$ neurons that project to the VTA to directly synergize with E2 mechanisms there that promote dopamine release during alcohol drinking. Future work will be able to parse the specific BNST$^{CRF}$ and other neuronal populations sensitive to E2 modulation utilizing strategies including neural activity monitoring with single-cell resolution. However, future technical advancement in the miniaturization and stabilization of endoscopes is

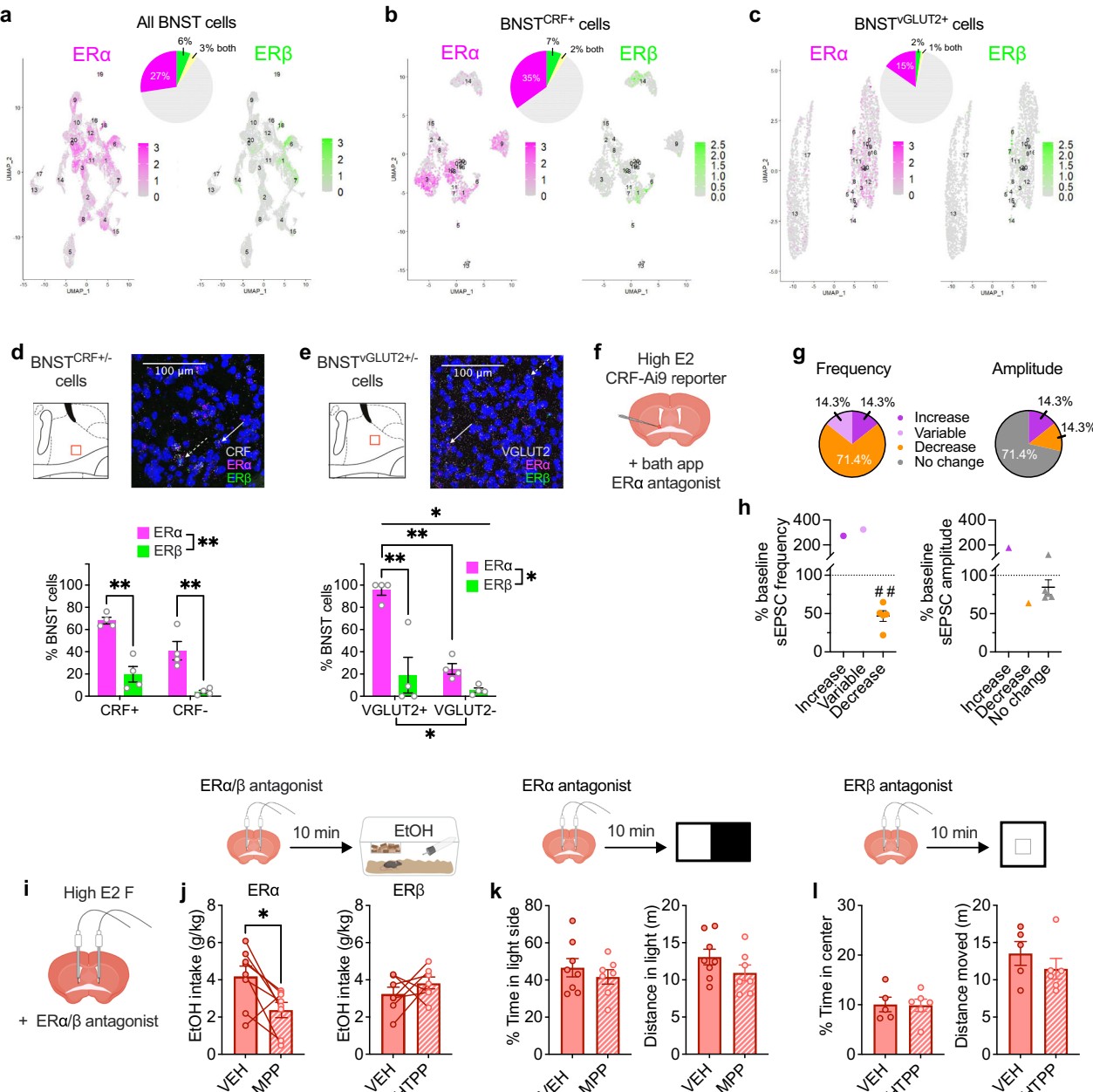

**Fig. 5 | ERα is robustly expressed in the BNST and mediates rapid E2 modulation of binge drinking but not avoidance behavior. a–c** Analysis of single nucleus RNA sequencing (snRNA-seq) of female mouse BNST nuclei (total cells: 38,806; GEO: GSE126836)[78]. **a** BNST cells expressing estrogen receptor α (ERα; *Esr1*) and estrogen receptor β (ERβ; *Esr2*) and both. **b** *Crh*-expressing BNST cells (BNST^CRF) expressing ERα, ERβ, and both. **c** *Slc17a6*-expressing BNST cells (BNST^VGLUT2) expressing ERα, ERβ, and both. **d**, **e** RNAscope fluorescence in situ hybridization (FISH) probing for ERα (*Esr1*), ERβ (*Esr2*), CRF (*Crh*), and VGLUT2 (*Slc17a6*) in the BNST in females, with red boxes indicating the location of confocal z-stack images taken; representative images pseudocolored for visibility. **d** ERα and ERβ expression in CRF+ and CRF- cells (N's = 4). Dashed arrow: cell expressing CRF/ERα/ERβ; solid arrow: cell expressing CRF/ERα. **e** ERα and ERβ expression in VGLUT2+ cells (N's = 4). Dashed arrow: cell expressing VGLUT2/ERα/ERβ; solid arrow: cell expressing VGLUT2/ERα. **f** Bath application of the ERα antagonist methyl-piperidino-pyrazole (MPP) on BNST^CRF neurons during slice electrophysiology recordings in high ovarian E2 female CRF-CrexAi9 reporters (Biorender license:

NF27GVWGQV). **g**, **h** Effects of bath application of MPP (3 µM) on spontaneous excitatory postsynaptic current (sEPSC) frequency and amplitude (N's = 4, 7 cells). **i** Depiction of strategy to site-deliver MPP (10 µM/200 nl/side), the ERβ antagonist 4-[2-phenyl-5,7-bis(trifluoromethyl)pyrazolo[1,5-a]pyrimidin-3-yl]phenol (PHTPP; 10 µM/200 nl/side), or saline (VEH) to the BNST in high E2 females (Biorender license: ZS27GW07FP). **j** Effects of intra-BNST MPP (N's=8, 8 VEH, 8 MPP) or PHTPP (N's = 7, 7 VEH, 7 PHTPP) on binge EtOH consumption (Biorender license: VB27JLIIQ2). **k** Effects of intra-BNST MPP on avoidance behavior in the light:dark box (LDB; N's = 8 VEH, 7 MPP). **l** Effects of intra-BNST PHTPP on avoidance behavior in the open field (OF; N's = 5 VEH, 6 PHTPP). *P < 0.05, **P < 0.01, ***P < 0.001 for 2xANOVA main effects and interactions between receptors; paired t-tests between VEH vs. ER antagonist treatment prior to DID; unpaired t-tests between VEH vs. ER antagonist treatment prior to avoidance assays; post hoc t-tests with H-S corrections as indicated. ##<0.01; one sample t-test. Data are presented as mean values +/−SEM. Detailed statistics are provided in Supplementary Table 1. Source data are provided as a Source Data file.

necessary to allow for long-term daily recording while maintaining normal estrous cyclicity and high levels of freely-moving voluntary binge drinking in female mice.

Intriguingly, while our data show that only rapid E2 signaling is needed for ovarian E2's role in alcohol drinking, they indicate that high ovarian E2-mediated anxiolysis instead may require E2-dependent transcriptional programming in the BNST and/or other brain regions. Consistent with another report[53], we found that acute modulation of E2 bioavailability did not alter avoidance behavior in intact females. And, there is ample evidence in the literature that ovariectomy increases avoidance behavior in female rodents[58,99]. However, the literature is mixed in the effects of E2 replacement, although only chronic administration has been evaluated[53,59,100]. Therefore, future work is needed to understand whether E2 is a critical mediator of estrous cycle modulation of avoidance behavior and to characterize the genomic effects by which it may act. Altogether, our results further our understanding of the mechanisms driving early alcohol drinking and states of anxiety in females, a critical step in understanding the role of a crucial neuromodulator and in developing effective interventions to minimize long-term negative health outcomes of alcohol drinking, especially in females.

## Methods

### Subjects
All experimental mice were male and female mice on a C57BL/6J background strain and ≥10 wks of age. Wildtype (WT) adult male and female C57BL/6J mice were purchased from Jackson Laboratory (Bar Harbor, ME, USA) at 8 weeks of age and all transgenic lines were bred in our animal facility. CRF-ires-Cre (CRF-Cre) [65]mice were bred with WT C57BL/6J mice, and hemizygous CRF-Cre mice were bred with homozygous floxed Ai9-tdTomato mice purchased from Jackson Laboratory to produce CRF-Cre-reporter mice. All mice were housed under a reverse circadian 12 h:12 h light:dark cycle with lights off at 7:30 am and *ad libitum* access to food and water, at ambient temperature (22–24 °C) and humidity (45%). Mice were singly housed at least one week prior to behavioral testing and remained singly housed throughout the experimental procedures. Experiments began approximately 3 h into the dark phase of the light:dark cycle. All procedures were conducted with approval of the Institutional Animal Care and Use Committee at Weill Cornell Medicine following the guidelines of the NIH Guide for the Care and Use of Laboratory Animals.

**Estrous cycle monitoring.** Estrous cycle stage was determined via daily lavage adapted from Mclean et al.[101] in which vaginal epithelial cells were collected with 15 μL 0.45% saline onto glass slides 1.5 h into the dark cycle and analyzed for cell types corresponding to the different estrous cycle stages (depicted in Fig. 1a).

**Tissue and plasma collection.** Naïve female mice estrous cycles were tracked daily for at least two weeks. Following prolonged tracking, mice categorized via cytology as being in either the proestrus (high E2 ovarian status) or metestrus (low E2 ovarian status) stages were rapidly decapitated under isoflurane anesthesia. Whole blood was collected from the trunk into tubes precoated with $K_2$EDTA (BD Microtainer). The whole blood was immediately spun down in a standard centrifuge at $1500 \times g$ for 10 min at 4 °C. The resulting plasma was collected and stored at −80 °C until further analysis. Simultaneously, ovaries were dissected, frozen immediately on dry ice, and stored at −80 °C.

### Estrogen concentration measurement
**Plasma.** 17β-Estradiol (E2) was quantified in pooled plasma samples by liquid chromatography-triple quadrupole tandem mass spectrometry (LC-MS/MS)[102]. Standard solutions of E2 (1.000 ± 0.005 mg/ml) and estradiol-2,4,16,16,17-d5 (E2-d5; 100 μg/ml) were obtained from Cerilliant (Round Rock, TX, USA) as 1 ml ampules with acetonitrile (ACN)

as solvent. Charcoal-stripped human serum (CSS; Mass Spect Gold MSG3100, Golden West Biologicals, Temecula, CA, USA) was spiked with a working E2 standard stock (2 μg/ml) using a Hamilton syringe to yield a high calibration standard with a final E2 concentration of 4 ng/ml. Two-fold serial dilutions of the high calibration standard were prepared in CSS to yield 14 calibration standards including a blank (0 ng/ml). Three in-house serum pools (2 human, 1 nonhuman primate) were used as quality control (QC) samples. Calibration standards, QC samples, and pooled plasma samples (150 μl) were reverse-pipetted into 350 μl V-bottom 96-well microtiter plates (Shimadzu, Kyoto, Japan) followed by the addition of 100 μl of an internal standard spike (3.7 ng/ml E2-d5) into each well. Plates were shaken on an orbital shaker at room temperature for 1 h and the contents of each well were transferred to 400 μl 96-well Isolute SLE+ plates (Biotage, Uppsala, Sweden) that had been washed with dichloromethane (DCM; 6 ×600 μl) and allowed to completely dry. Plates were equilibrated at room temperature for five minutes then eluted with DCM (3 ×600 μl) into 2 ml 96-well round bottom polypropylene plates (Analytical Sales & Services, Flanders, NJ, USA) containing 100 μl of 2-propanol per well. Plates were then subjected to positive pressure (-5 psi, 60 s) using a Pressure+ 96 positive pressure manifold (Biotage) to elute residual eluent. Eluates were evaporated to dryness under nitrogen at 40 °C using a TurboVap 96 evaporation system (Biotage) and samples were reconstituted in 50 μL of 25:75 methanol:water (v/v) and transferred to new 350 μl V-bottom 96-well microtiter plates for analysis.

LC-MS/MS analysis was performed on a Shimadzu LCMS-8060 system. Twenty-five microliters of each reconstituted sample were injected from the microtiter plate (held at 10 °C) onto a Raptor Biphenyl column (100 mm × 2.1 mm × 2.7 μm particle size; Restek, Bellefonte, PA, USA) with guard cartridge (5 mm × 2.1 mm × 2.7 μm particle size; Restek) held at 35 °C. Analytes were eluted with an eluent gradient initially consisting of 35% mobile phase A (0.15 mM ammonium fluoride in water) and 65% mobile phase B (pure methanol) that was linearly changed to 97% B over 4 min and then to 100% B over 1.1 min. The column was then washed with 100% B for 1.8 min before being returned to 65% B over 0.1 min and allowed to equilibrate for 3.4 min, for a total method time of 10.4 min. The flow rate was held constant at 0.4 ml/min. Analytes were detected via heated electrospray ionization in negative ionization mode with scheduled multiple reaction monitoring (E2: 271.05 > 145.05 quantifier, >143.05 reference; E2-d5: 276.20 > 187.10 quantifier, >147.10 reference). Heating gas (air; 10 L/min), drying gas (nitrogen; 10 L/min), and nebulizing gas (nitrogen; 3 L/min) were provided by a Peak Genius 1051 gas generator (Peak Scientific, Inchinnan, Scotland, UK). The interface temperature was 300 °C, the heat block temperature was 400 °C, and the desolvation line temperature was 250 °C. Argon (Airgas) at 320 kPa was used as the collision gas. The source was equipped with a capillary B needle (Shimadzu) and capillary protrusion and probe distance were determined empirically based on maximizing E2 response. Shimadzu Lab Solutions software (version 5.97) was used to optimize the quadrupole pre-bias voltages and collision energies; the interface voltage −2 kV. The calibration curve was constructed using linear regression with 1/C weighting of the analyte/internal standard peak area ratio. The lower limit of quantification (LLOQ) for E2 was 1 pg/ml, defined as the lowest calibration curve concentration having an average accuracy of 80–120% and a CV of <20%[103].

**Brain.** Frozen brains (~450 mg) were sliced into ~50–75 mg sections and added to glass test tubes (12 ×75 mm) with 2 ml of cold (−20 °C) acetonitrile and 50 μl of internal standard (14.8 ng/ml E2-d5 in 3:1 water:methanol, v:v). Samples were homogenized using a handheld homogenizer (Tissue Master 125, Omni International, Kennesaw, GA, USA) until a homogenous suspension formed. The homogenizer bit was then rinsed in 1 ml of acetonitrile in a separate tube and this solution was combined with the homogenate. The homogenizer was

cleaned between samples by rinsing consecutively in methanol, 2-propanol, and water, followed by a final rinse in fresh methanol. Homogenates were stored at −20 °C until further processing, at which point they were centrifuged and the supernatants transferred to new test tubes and dried under forced air at 37 °C. Samples were resuspended in 200 μl of 3:1 water:methanol (v:v) and subjected to supported liquid extraction and analysis of E2 by LC-MS/MS as described for plasma samples except that the calibration standards (200 μl) were prepared in 3:1 water:methanol (v:v) and 50 μl of internal standard (14.8 ng/ml E2-d5 in 3:1 water:methanol, v:v) were used.

**Western immunoblotting.** Tissue was homogenized using a Teflon pestle (Pyrex, Illinois, USA) in cold RIPA buffer (Pierce, Massachusetts, USA) containing protease and phosphatase inhibitors. After spinning the homogenate at $12,000 \times g$ for 20 min, the supernatant was collected and used for western immunoblotting. Twenty μg of protein per sample were separated by 10% SDS-PAGE and transferred to polyvinylidene difluoride membranes, which were blocked in 5% nonfat dry milk and incubated 24–73 h at 4 °C in a primary antibody prior to being incubated in peroxidase-linked IgG conjugated secondary antibody at 1:5000 at room temperature for 1 h. Blots were incubated with aromatase (PA1-16532; Invitrogen, California, USA) diluted to 1:1000 and GAPDH (ab22555l; Abcam, Cambridge, UK) diluted to 1:40000. Proteins were quantified at the following molecular weights: aromatase at 55 kDa and GAPDH at 36 kDa. Protein bands were quantified as mean optical intensity using ImageJ software (NIH) and normalized to GAPDH. Normalized optical density values from each sample were used to calculate the fold change for each proestrus group compared with the metestrus group (set to 1.0) on the same blot.

**Single nucleus RNA sequencing.** Gene-expression matrices from female mouse BNST samples from dataset GEO: GSE126836[78] were loaded into R (v 4.3.1)[104] and preprocessed using Seurat[105]. We normalized the data using *NormalizeData*. Variable features were found using *FindVariableFeatures* using the vst selection method and nfeatures = 2000. Cells expressing genes of interest were subsetted and clustered. UMAPs showing co-expression of genes were generated with *FeaturePlot*.

**RNAscope fluorescence in situ hybridization.** Mice were euthanized under deep isoflurane anesthesia and their brains rapidly extracted, flash frozen in −35 °C 2-methylbutane, and stored at −80 °C under RNase-free conditions. Brains were embedded in OCT blocks with four brains per block, and 20 μM coronal sections were sliced on a cryostat and collected onto Superfrost Plus slides and stored at −80 °C in sealed slide boxes until undergoing in situ hybridization. Slides were thawed and in situ hybridization was conducted per manufacturer instructions using RNAscope Multiplex Fluorescent v1 kit with the HybEZ Oven and the following ACD probes: VGLUT2: mm-Slc17a6, cat #319171; CRF: mm-Crh, cat #316091; ERa: mm-Esr1, cat #478201; ERb: mm-Esr2, cat #316121. Slides were counterstained with DAPI and cover slipped with Vectashield. Images were acquired on a Zeiss LSM 880 Laser Scanning Confocal microscope in 1 μM plane z-stacks and manually analyzed in Metamorph according to RNAscope instructions to quantify the number of DAPI+ nuclei and number of DAPI+ cells expressing each probe alone or in combination. Representative images (Fig. 5d, e) were pseudocolored following analysis for color matching purposes using ImageJ (1.53a).

**Drugs.** Cyclodextrin-encapsulated E2 (Millipore Sigma, Massachusetts, USA) was dissolved in 0.9% saline, letrozole (LET; Tocris Bioscience, Bristol, UK) was dissolved in 5% dimethyl sulfoxide in 0.9% saline. MPP dichloride and PHTPP (Tocris Bioscience) were dissolved in 1% dimethyl sulfoxide and 0.9% saline. Membrane-only E2 (E2-BSA; Steraloids, Rhode Island, USA) was purified using Zeba™ Spin Desalting Columns,

7 K (Thermo-Fischer, Massachusetts, USA) as follows: the column's storage solution was removed by centrifugal spinning at $1500 \times g$ for 1 min, then the column was conditioned with phosphate buffered saline ($300 \, \mu l \times 3$). Membrane-only E2 (approx. 2 mg/min in phosphate buffered saline, 120 μl) was loaded and eluted by centrifugal spinning at $1500 \times g$ for 2 min. The exact concentration (36 μM) was assessed using BCA protein assay.

**Stereotaxic surgery.** For experiments requiring site-directed administration of viral vectors or cannulae implantation, mice were anesthetized with 2% isoflurane (VetEquip, CA, USA) in 0.8% oxygen in an induction chamber (VetEquip,) then placed in an Angle Two mouse stereotaxic frame (Leica Biosystems, Wetzlar, Germany) and secured with ear bars into a nose cone delivering isoflurane to maintain anesthesia. Mice were given a subcutaneous injection of meloxicam (2 mg/kg) for preemptive analgesia and 0.1 mL of 0.25% Marcaine around the incision site. For administration of viral vectors, a Neuros 7000 series 1 μL Hamilton syringe with a 33-gauge needle (Reno, NV) connected to a remote automated microinfusion pump (KD Scientific, Holliston, MA) was used for construct delivery at a rate of 50–100 nL/min to the BNST (A/P: + 0.3 mm, M/L: ±1.1 mm, D/V: − 4.35 mm, 250 nL). Following infusion, the needle was left in place for 10 min and then slowly manually retracted to allow for diffusion and prevent backflow of the virus. Mice were continuously monitored for at least 30 min post-surgery to ensure recovery of normal breathing pattern and sternal recumbency and then checked daily. For cannulae implantation surgeries, bilateral cannulae (P Technologies, Virginia, USA) cut to 2.2 mm below the pedestal were secured with C & B Metabond dental cement at the same BNST coordinates used for viral infusions. The cannulae were secured with an internal dummy and plastic dust cap (P Technologies). For cannula histologies, mice were deeply anesthetized with isoflurane and 200 nL dye (Chicago Sky, Millipore Sigma) was infused. Brains were rapidly extracted from anesthetized animals and flash frozen, embedded in OTC, and sectioned at 45 μm on the cryostat to ensure placement of cannula. DREADD and fiber photometry brains were extracted, post-fixed overnight in 4% PFA, and then placed in PBS until they were sliced on the coronal plane in 45 μm sections on a VT1000S vibratome (Leica Biosystems, Wetzlar, Germany) to check injection placements and viral expression.

**Behavior assays**

**Drinking in the Dark.** The Drinking in the Dark (DID)[50] binge alcohol drinking paradigm was used to model human binge consumption behavior in mice. On Days 1-3 of each cycle of alcohol DID, three hours into the dark cycle, the home cage water bottle was replaced with a bottle containing 20% (v/v) ethanol for two hours and four hours on Day 4, followed by three days of abstinence between cycles. The same schedule was used to evaluate binge sucrose consumption, except that home cage water bottles were replaced with 1% (w/v) sucrose. For all drinking experiments, empty "dummy" cages on the same rack as behavior mice received the same alcohol or sucrose bottle replacement, and consumption was adjusted for a leak from dummy bottles and then normalized to bodyweight.

**Avoidance behavior.** Avoidance behavior was assessed with a battery of assays including the open field test, elevated plus maze, and light:dark box[52,65]. SMART software (Panlab, Barcelona, Spain) and Ethovision 15 (Noldus, Wageningen, Netherlands) video tracking were used to quantify raw locomotor and location data used to calculate measures including distance traveled and time spent. For basal avoidance behavior (Fig. 1j–n), assays were performed once per week within the same cohorts of mice.

**Open field test.** The open field test (OF) was conducted in a novel $50 \times 50$ cm plexiglass arena. Mice were placed in a corner of the arena,

and time and distance in the center 50% of the apparatus vs. corners and periphery across the 30 min trial were calculated.

**Elevated plus maze.** The elevated plus maze (EPM) was conducted in a plexiglass maze with two open and two closed arms (35 cm length × 5.5 cm width, with walls 15 cm tall in the closed arms). Mice were placed in the center of the EPM and time and distance in the open arms, closed arms, and center platform were calculated for the five minute trial.

**Light:Dark Box.** The light:dark box assay (LDB) was conducted in a rectangular box divided into two equal compartments (20 cm l × 40 cm w × 34.5 cm h) with one dark with a closed lid and the other with an open top and illuminated by two 60 W bulbs placed 30 cm above the box. The two compartments were separated by a divider with a 6 cm × 6 cm cut-out passageway at floor level. The mouse was placed in a corner of the light compartment and allowed to freely explore the apparatus, and the number of light side entries and time spent in the light vs. dark compartment were calculated for the 10 min trial.

**Pharmacological manipulations.** Systemic LET (1 or 10 mg/kg[106,107]) or vehicle control were administered via intraperitoneal (i.p.) injection 40 min prior to alcohol or sucrose bottles being placed in the cages. For central infusions via cannulation, cyclodextrin-encapsulated E2 (20 pg), membrane-only E2 (55 pg), MPP (10 μM), or PHTPP (10 μM) in a total volume of 200 nL was infused bilaterally into BNST targeted guide cannulae using two Neuros 7000 series 1 μL Hamilton syringes with 33-gauge needles attached to a Harvard Apparatus pump. Ten minutes following BNST infusion, DID ethanol or sucrose bottles were placed into cages for DID or mice were placed in the testing apparatuses for avoidance assays.

**Chemogenetic manipulations.** CRF-Cre mice were injected with a cocktail of viral vectors containing Cre-dependent excitatory and inhibitory Designer Receptors Exclusively Activated only by Designer Drugs (DREADDs) (1:1 cocktail of the excitatory Gq-DREADD AAV2-hSyn-DIO-hM3D[Gq]-mCherry [125 nL] plus inhibitory Gi-KORD AAV8-hSyn-DIO-KORD[Gi]-mCitrine [125 nL]) or control virus (AAV2-hSyn-DIO-mCherry [250 nL]) in the BNST. After three weeks to allow for viral expression, mice underwent behavioral tests (conducted under 200–250 lux lighting conditions) to evaluate the role of the BNST^CRF neurons in mediating avoidance behaviors and alcohol DID, All behavioral experiments/drinking sessions commenced three hours into the dark cycle, and each avoidance behavioral apparatus was thoroughly cleaned with 70% ethanol before each trial. For chemogenetic manipulation experiments, mice received vehicle (0.9% sterile saline, i.p.) or DREADD ligand dissolved in vehicle (CNO, 5 mg/kg, i.p., or SalB, 10–17 mg/kg, i.p.) injections 10–40 min prior to behavioral testing. Estrous cycle status was not monitored in DREADD behavioral experiments (EtOH DID, anxiety measures). Following the experiments, histological verification was performed on each mouse to verify virus expression.

**Fiber photometry.** CRF-Cre mice expressing GCaMP6s in the BNST and a permanent fiber cannula implanted above the BNST were first habituated to handling and tethering of fiber optic patch cables during the week prior to the avoidance behavioral assays. Optical patch cables were photobleached through the fiber photometry workstation (Tucker-Davis Technologies, Inc., Alachua, Florida) for at least 12 h prior to being tethered to the mouse's head stage prior to each behavioral session. For drinking experiments, mice underwent a modified DID procedure in which they received water access prior to and after each 2-h alcohol access. Drinking was measured using a capacitance based lickometer that elicited a TTL signal when licked, and a drinking bout was defined as at least 2 TTL pulses within one s,

with a minimum bout length of 0.5 s. Raw fiber photometry signals (465 nm channel for GCaMP, 405 nm channel for the isosbestic point control) were collected at a sampling rate of 1017.25 Hz and denoised using a median filter to remove electrical artifacts and a lowpass filter to reduce high-frequency noise. To eliminate noise in the signal due to transfer to the behavioral apparatus or associated with bottle changes, the first 150 s for EPM, 400 s excluded for OF, and 1000 s for DID were excluded from analysis. The EPM trial length was increased to provide 5 min of analyzed data. The preprocessed data were down sampled by factor of 100 and corrected for photobleaching and motion artifacts by finding the best linear fit of the isosbestic control signal to the GCaMP signal and subtracting this estimated motion component from the GCaMP signal. Final z-score values were calculated by subtracting the mean of the whole trace from each datapoint and dividing it by the standard deviation of the trace. Data were analyzed based on behavioral epoch or tracked location. For EtOH DID, Fiber traces were segmented via a custom Python script from Jupyter open-source software to extract the last 30 min of W1, the first and last 30 min of EtOH, and the first 30 min of W2 as CSV files for each corresponding animal. Each file was converted into an Axon Binary File (integer)(*.abf) through the Clampfit 10.0 software. Peak analysis was executed in the Mini Analysis program Version 6. 0. 3 (Synaptosoft, USA). Detection parameters were adjusted per trace to automatically detect peaks that had a high ratio of signal to noise. Equal detection parameters were applied to each set of W1, EtOH, and W2 traces, grouped by the corresponding DID day. Each trace was manually checked by a blinded experimenter to eliminate false positives. Amplitude and frequency were binned and measured via Excel, and Graph Pad Prism 10. Following the experiments, histological verification was performed on each mouse to verify virus expression.

**Ex vivo slice electrophysiology.** CRF-Cre reporter mice were rapidly decapitated under isoflurane anesthesia and their brains extracted. Brains were blocked on the coronal plane and acute 300 μm coronal slices prepared and incubated in carbogenated solutions with a pH of 7.35 and osmolarity of 305[50]. Sections including BNST were sliced in N-methyl-D-glucamine (NMDG) artificial cerebrospinal fluid (NMDG-aCSF) at RT containing (in mM): 92 NMDG, 2.5 KCl, 1.25 $NaH_2PO_4$, 30 $NaHCO_3$, 20 HEPES, 25 glucose, 2 thiourea, 5 Na-ascorbate, 3 Na-pyruvate, 0.5 $CaCl_2 \cdot 2H_2O$, and 10 $MgSO_4 \cdot 7H_2O$. Slices were transferred to NMDG-aCSF at 32 °C for 12–14 min, and then incubated for at least 1 h at RT in HEPES-aCSF containing (in mM): 92 NaCl, 2.5 KCl, 1.25 $NaH_2PO_4$, 30 $NaHCO_3$, 20 HEPES, 25 glucose, 2 thiourea, 5 Na-ascorbate, 3 Na-pyruvate, 2 $CaCl_2 \cdot 2H_2O$, and 2 $MgSO_4 \cdot 7H_2O$. Slices were placed in the recording chamber and perfused at a rate of 2 ml/min with 30 °C normal aCSF containing (in mM): 124 NaCl, 2.5 KCl, 1.25 $NaH_2PO_4$, 24 $NaHCO_3$, 12.5 glucose, 5 HEPES, 2 $CaCl_2 \cdot 2H2O$, and 2 $MgSO_4 \cdot 7H_2O$, for at least 20 min prior to electrophysiological recordings. CRF+ (tdTomato-tagged) neurons were visualized using a 560 nm LED and 40× immersed objective with DsRed filter14. Basal synaptic transmission was measured in the voltage-clamp configuration using a cesium-methanesulfonate-based intracellular recording solution containing (in mM): 135 CsMeth, 10 KCl, 10 HEPES, 1 MgCl2·6H2O, 0.2 EGTA, 4 Mg-ATP, 0.3 Na2GTP, 20 phosphocreatine (pH 7.3, 290 mOsm) and driving force (−55 mV and +10 mV) to isolate excitatory and inhibitory transmission within individual neurons. The effects of E2 pharmacology on spontaneous excitatory postsynaptic currents (sEPSCs, 2 cells/mouse) were measured in a voltage clamp configuration using a cesium-gluconate-based intracellular recording solution containing (in mM): 117 D-gluconic acid, 20 HEPES, 0.4 EGTA, 5 TEA, 2 MgCl2·6H2O, 4 Na-ATP and 0.4 Na-GTP (pH 7.3 and 290 mOsm),) and a holding potential of -70 mV, using picrotoxin (25 μM) in the bath to block GABAergic synaptic transmission. E2 (1 nM, 10 nM, 100 nM, 1000 nM), mE2 (100 nM), or MPP (3 μM) in aCSF were washed on to measure their effects on excitatory synaptic transmission in separate

experiments. Responder category threshold was defined as 15% delta of average drug wash on period OR 50% delta in a one minute period during drug wash on. Signals were acquired using a Multiclamp 700B amplifier (Molecular Devices), digitized, and analyzed via pClamp 10.4 or 11 software (Molecular Devices). Input resistance and access resistance were continuously monitored throughout experiments, and cells in which properties changed by more than 20% were not included in data analysis.

**Schematic creation**. Illustrations were created with biorender.com.

**Statistical analyses**. Statistical analyses were performed in GraphPad Prism 10. Data in each group for all dependent measures were checked for their distributions in raw and log space within the group and equality of variance across groups and analyzed accordingly. Statistical comparisons were always performed with an alpha level of 0.05 and using two-tailed analyses. Paired t-tests for within-subjects and unpaired t-tests for between-subjects direct comparisons were used to evaluate the effects of E2 modulators on alcohol drinking and avoidance behaviors. Two-way and three-way repeated measures ANOVAs (RM-ANOVAs) were used to examine the effects of cycle, groups, and drug and other repeated variables between mice in different experimental conditions; mixed effects models were used when one or more matched data point was unavailable. Following significant effects in ANOVAs, *post hoc* direct comparisons were performed using unpaired or paired t-tests with Holm–Sidak (H-S) correction for multiple comparisons, and adjusted *p* values are reported. Linear regressions were performed for correlation analyses. Data in figures are presented as mean ± SEM, and raw data points are included in all figures except some repeated measures graphs in which there were too many raw data points to be clearly represented. All statistics depicted in Supplementary Table 1.

## Data availability
The raw data used in this study are available in the Figshare database (https://figshare.com/projects/Rapid_nongenomic_estrogen_signaling_controls_alcohol_drinking_behavior_in_mice/225168). Source data are provided with this paper as a Source Data file. Source data are provided with this paper.

## Code availability
All custom code written in R and Python have been deposited on GitHub (https://github.com/kpleil/Zallar-et-al.-2024).

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

## Acknowledgements
National Institutes of Health grant K99/R00 AA023559 (K.E.P.). National Institutes of Health grant R01 AA027645 (K.E.P.). NARSAD Young Investigator Award (K.E.P.). Stephen and Anna-Maria Kellen Foundation Junior Faculty Award 26608 (K.E.P.). National Institutes of Health grant F31 AA029293 (L.J.Z.). National Institutes of Health grant T32 DA039080 (J.K.R.I., L.J.Z., O.B.L.). National Institute of General Medical Sciences of the National Institutes of Health Medical Scientist Training Program grant to the Weill Cornell/Rockefeller/Sloan Kettering Tri- Institutional MD-PhD Program T32GM007739 (C.K.H.). Deutsche Forschungsgemeinschaft Walter Benjamin Fellowship 516215326 (P.B.). National Institutes of Health grant P51 OD011092 (K.M.F., D.W.E.). National Institutes of Health grant T32 GM141949 (I.P.).

## Author contributions
Conceptualization: K.E.P. Methodology: K.E.P., L.J.Z. Investigation: L.J.Z., J.K.R.I., P.U.H., I.P., D.L., J.P.W., A.S.R.R., R.B., J.A., O.B.L., M.J.S., C.K.H., K.M.F., H.M., S.N., J.M., P.G., P.B., A.S., L.V.H., D.W.E. Visualization: L.J.Z., J.K.R.I., P.U.H., K.E.P. Funding acquisition: K.E.P., L.J.Z., J.K.R.I., O.B.L., C.K.H., P.B., K.M.F., D.W.E. Project administration: K.E.P. Supervision: K.E.P., D.W.E., J.G. Writing – original draft: KEP, L.J.Z. Writing – review & editing: L.J.Z., J.K.R.I., P.U.H., I.P., D.L., J.P.W., A.S.R.R., R.B., J.A., O.B.L., M.J.S., C.K.H., K.M.F., H.M., S.N., J.M., P.G., P.B., A.S., L.V.H., D.W.E., J.G., K.E.P.

## Competing interests
The authors declare no competing interests.
