## [Transparent Peer Review file · Nature Communications]

Rapid nongenomic estrogen signaling controls alcohol drinking behavior in mice

Corresponding Author: Dr Kristen Pleil

Version 0:

Reviewer comments:

Reviewer #2

(Remarks to the Author)

The authors have addressed my main concerns in the revised manuscript with the addition of experiments that demonstrate that intra-BNST infusion of letrozole in female mice does not affect ethanol consumption and that systemic injection of letrozole in male mice does not affect ethanol consumption. These results support their conclusions that ovarian-derived E2 promotes binge ethanol consumption through rapid signaling mechanisms in the BNST. I also appreciate the other changes that the authors made referencing additional papers and more precisely stating the conclusions and implications of their findings. My only request is that the experiment with systemic letrozole treatment in male mice be added to Extended data Fig. 5 (related to Fig. 4).

(Remarks on code availability)

Reviewer #3

(Remarks to the Author)

In the manuscript by Zallar et al., the authors present original and novel results. There is a large amount of work showing that females have elevated alcohol intake and that estrogen is involved in this effect. This study, however, is the first to strategically monitor, classify these estrogen states (low, high), and then investigate the brain region (BNST), cell type (CRF), and signaling cascades associated (ER α , non-genomic mechanism). Understanding these sex effects in alcohol consumption are critically important to the field and this work will help guide future work in this area. The experiments have been well-designed and appropriately analyzed. The authors were very responsive to the original reviews and addressed the previous concerns with new experiments, reduced emphasis on the DREADD experiments, and re-framing of interpretations. These changes have greatly improved the manuscript and there are no further concerns.

(Remarks on code availability)
